# Elements of Sleep Breathing and Sleep-Deprivation Physiology in the Context of Athletic Performance

**DOI:** 10.3390/jpm12030383

**Published:** 2022-03-02

**Authors:** Dimitra D. Papanikolaou, Kyriaki Astara, George D. Vavougios, Zoe Daniil, Konstantinos I. Gourgoulianis, Vasileios T. Stavrou

**Affiliations:** 1Laboratory of Cardio-Pulmonary Testing and Pulmonary Rehabilitation, Department of Respiratory Medicine, Faculty of Medicine, University of Thessaly, 41110 Larissa, Greece; dimitra.papanikolaoy2001@gmail.com (D.D.P.); kyriakiastarara@gmail.com (K.A.); dantevavougios@hotmail.com (G.D.V.); zdaniil@uth.gr (Z.D.); kgourg@uth.gr (K.I.G.); 2Department of Neurology, 417 Army Equity Fund Hospital (NIMTS), 11521 Athens, Greece; 3Department of Neurology, Faculty of Medicine, University of Cyprus, Nicosia 2029, Cyprus

**Keywords:** sleep deprivation, exercise, cardiovascular, cognitive

## Abstract

This review summarizes sleep deprivation, breathing regulation during sleep, and the outcomes of its destabilization. Breathing as an automatically regulated task consists of different basic anatomic and physiological parts. As the human body goes through the different stages of sleep, physiological changes in the breathing mechanism are present. Sleep disorders, such as obstructive sleep apnea-hypopnea syndrome, are often associated with sleep-disordered breathing and sleep deprivation. Hypoxia and hypercapnia coexist with lack of sleep and undermine multiple functions of the body (e.g., cardiovascular system, cognition, immunity). Among the general population, athletes suffer from these consequences more during their performance. This concept supports the beneficial restorative effects of a good sleeping pattern.

## 1. Sleep-Disordered Breathing Physiology

### 1.1. Respiratory Aspect

The mechanism of breathing includes air flow through the passages of the respiratory system due to pressure gradients that are formed by contraction of the diaphragm and the thoracic muscles. Air flows from a region of higher pressure to a region of lower pressure. Respiration involves the interplay between three different pressures: the atmospheric, the interalveolar, and the intrapleural pressure. Inspiration is the active phase of respiration and the result of muscle contraction, and expiration is the passive phase in calm state. Regulation of respiratory system is subconscious and determines rhythmic rotation between inspiration and expiration and ventilation (breathing frequency and depth) [1].

Sleep state is associated with significant changes in respiratory physiology, including ventilatory responses to hypoxia and hypercapnia, upper airway, and intercostal muscle tone, and tidal volume and minute ventilation. These changes are further magnified in certain disease states, such as chronic obstructive pulmonary disease, restrictive respiratory disorders, neuromuscular conditions, and cardiac diseases [2]. Sleep-disordered breathing (SDB), which causes sleep deprivation and intermittent hypoxia, encompasses a broad spectrum of sleep-related breathing disorders, including obstructive sleep apnea (OSA), central sleep apnea (CSA), as well as sleep-related hypoventilation and hypoxemia. Relative hypotonia of respiratory muscles, body posture changes, and altered ventilatory control result in additional physiologic changes contributing to hypoventilation [3]. Hypercapnia, hypoxemia, and negative intrathoracic pressure swings lead to increased sympathetic response in order to maintain the normal air flow followed by hyperventilation.

### 1.2. Neural Aspect

Breathing is an automatic function and is regulated, according to the metabolic demands, by the autonomic nervous system (ANS) and, more specifically, by the respiratory center (RC), a central pattern generator (CPG) located in medulla oblongata along with the other vital reflexes. Cortical–medullary circuits furthermore guarantee that voluntary control of breathing is possible [4]. Upon loss of cortical functions without the loss of the medullary CPG, however, control is maintained by the latter.

### 1.3. Input Sensors

Wakefulness, non-rapid eye-movement sleep (NREM), and rapid eye-movement sleep (REM) sleep represent three distinct states during the sleep–wake cycle [5]. Breathing is maintained during sleep, but its regulation differs from wakefulness [6]. The progression through sleep stages is accompanied by a sequence of physiological changes based on chemoreceptor and baroreceptor reflexes [7]. Chemoreceptors are divided into peripheral and central. Chemoreflex input consist of peripheral (carotid and aortic bodies), which reflect the concentrations of arterial O_2_, and of central receptors, which are sensitive to CO_2_ and H^+^ changes in the CSF [8]. Consequently, the ventilatory feedback control system of the chemoreflex is vulnerable to rapid fluctuations of this input, similar to those that occur during NREM sleep [9].

Two additional respiratory control centers exist in the medulla: the vasomotor (VMC) that regulates blood pressure and the cardiac center (cardioinhibitory and cardioacceleratory centers) for the regulation of heart rate. The three centers are interconnected to function coordinately for the release not only of the chemoreflex but also for the baroreflex [10]. The baroreceptor reflex is activated when blood pressure is found increased by the baroreceptors in walls of carotid internal artery and of aorta and vasodilation occurs (inhibition of VMC) as well as decreased heart rate (stimulation of cardioinhibitory centers).

In sleep-disordered breathing, the circle of intermittent hypoxia–hypercapnia stimulates chemoreflex entirely, which in turn overstimulates SNS, attenuates baroreflex, and enhances hyperventilation after arousal [11]. Arousal occurs in order to increase the muscle tone and compensate for hypoventilation. Interestingly, the increased tone of SNS persists during daytime, too. As baroreflex is desensitized, the PNS is incapable of antagonizing the detrimental effects of SNS overstimulation, demonstrating mainly hypertension and tachycardia.

### 1.4. Output Mediators

The mutable environment of respiratory regulation during sleep affects multiple systems and structures: the ANS as well as lungs, chest wall, and upper airway [12]. During wakefulness and REM, sympathetic tone is dominant, whereas during NREM sleep, parasympathetic tone prevails to create a state of reduced activity [13]. Therefore, blood pressure and heart rate are reduced during NREM, whereas in REM sleep, the pulses of sympathetic activity induce tachycardia and relatively increase blood pressure [14].

During sleep, ventilation and functional residual capacity decrease slightly [15]. In stage I of NREM sleep, sufficient muscle tone is maintained, and frequent body posture changes occur. Respiratory pattern is more regular, while minute ventilation is progressively reduced, resulting in an increase of end-tidal carbon dioxide (ETCO_2_) compared to a waking state. During REM, respiratory pattern varies while ventilation further drops, accompanied by a slight reduction in oxygen saturation [16].

These fluctuations of arterial blood pressure, heart rate, and respiration occur in NREM and REM sleep and during transitions between sleep and arousal [17]; they may explain the sensitivity differences in hypoxia–hypercapnia, a major pathophysiologic element in sleep-disordered breathing [5]. Pulmonary stretch receptors work in coordination with central and peripheral chemoreceptors as the corresponding reflexes affect upper airway and respiratory pump muscles. The relationship is displayed in detail in Figure 1. A reduction in respiratory muscle tone occurs during NREM sleep but is more prominent during REM [18], attenuating the occlusion pressure responses to both hypoxia and hypercapnia in REM sleep stage, a clinical phenomenon consistent with emerging even in normal people [19]. In this context, arousals emerge, fragmenting sleep architecture. A protective reflex is activated by local upper airway (UA) mechanoreceptors due to the negative pressure in the UA, preventing its collapse by enhancing activity of UA dilators [20]. This reflex re-establishes ventilation in an alternative-to-arousal manner.

## 2. Sleep Deprivation

Sleep-disordered breathing is associated with sleep deprivation. This sleep disruption interferes with the normal restorative functions of NREM and REM sleep, resulting in disruptions of breathing and cardiovascular function, changes in emotional reactivity, and cognitive decline in attention, memory, and decision making [21]. Sleep-disordered breathing is common among overweight and obese children. It is a risk factor for several health complications, including cardiovascular disease. Inflammatory processes leading to endothelial dysfunction are a possible mechanism linking SDB and cardiovascular disease [22,23].

### 2.1. Sleep Deprivation and CO_2_ Retention

Disordered breathing is commonly associated with hypercapnia, which is followed by sufficient CO_2_ retention. This phenomenon leads to various impairments due to dangerous levels of hypercapnia. Acute responses to CO_2_ affect breathing primarily via central chemoreceptors [24]. Retention of CO_2_ not only contributes to chemoreflex via hypercapnia and acidosis but also serves as a powerful stimulus to increase respiration. Hypoxia potentiates the effects of CO_2_, resulting in a stronger ventilatory response. Through various mechanisms, retention of CO_2_ can persist during daytime, too [25].

Carbon dioxide retention is related to oxidative stress and increased sympathetic activity with subsequent effects, such as hypertension. Recent evidence has now implicated a role for oxidative stress in sleep and sleep loss [26]. Oxidative stress is defined by increased oxygen reactive species (ROS) production and inability of the cell to alienate them. Prolonged wakefulness/sleep deprivation activates an adaptive stress pathway termed the unfolded protein response, which temporarily guards against the deleterious consequences of reactive oxygen species [24,26]. The elevated sympathetic response also triggers a generalized inflammatory cascade that is associated with the pathophysiology of multiple comorbidities, including insulin resistance, hypertension, diabetes, atherosclerosis, and metabolic syndrome [27]. Epidemiologic studies in adults and children and laboratory studies in young adults indicate that sleep deprivation may be associated with several relevant impairments: decreased glucose tolerance, decreased insulin sensitivity, increased evening concentrations of cortisol, increased levels of ghrelin, decreased levels of leptin, and increased hunger and appetite (Table 1). Nevertheless, the current epidemic of obesity could be partly attenuated by better sleep regulation [28]. In healthy adults who are chronically sleep restricted, a simple, low-cost intervention, such as sleep extension, is feasible and is associated with improvements in fasting insulin sensitivity [29]. In the matter of inflammatory system, sleep loss triggers signaling pathways in the brain and periphery. The Toll-like receptor 4 (TLR4) activates inflammatory signaling cascades in response to endogenous and pathogen-associated ligands known to be elevated in association with sleep deprivation. TLR4 is therefore a possible mediator of some of the inflammation-related effects of sleep loss [30]. Furthermore, total sleep loss produces significant increases in plasma levels of sTNF-alpha receptor I and IL-6, messengers that connect the nervous, endocrine, and immune systems [31].

### 2.2. Sleep Deprivation and Exercise: Cognitive Implications

A sleep-deprived brain fails to recuperate neurons, undermining cognitive performance. General cognitive assessment tests unveil the cognitive phenotype of SD, especially in attention and short-term memory, as they anatomically overlap [36,37]. Furthermore, SD, in the context of sleep apnea, affects learning and memory [38,39] (Table 1). Furthermore, other daytime consequences, such as excessive sleepiness and fatigue, coexist and interact with cognitive impairment [40]. These are linked with various effects on exercise, including athletic performance, reaction time, accuracy, strength and endurance [41]. Alertness, judgment, and decision making suffer due to SD, shifting motivational behaviors towards sleep-promoting goals [42,43].

Sleep deprivation of 30 to 72 h consecutively does not affect cardiovascular and respiratory responses to exercise of varying intensity or the aerobic and anaerobic performance capability of individuals. Muscle strength and electromechanical responses are also not affected. Time to exhaustion, however, is decreased by sleep deprivation [44]. Research indicates that some maximal physical efforts and gross motor performances can be maintained. Effects on cognitive function consist of slower and less accurate cognitive performance. Reduction in sleep quality and quantity could result in an autonomic nervous system imbalance, simulating symptoms of the overtraining syndrome [45]. The integrity of sleep architecture seems to determine subjective sleep quality and waking performance. The effects of insufficient sleep primarily concern subjective and objective sleepiness as well as attention, whereas performance on higher cognitive functions appears to be better preserved albeit at the cost of increased effort [46]. All in all, sleep deprivation induces a vulnerability in various domains of cognition, leading to overall suboptimal performance.

This vulnerability to cognitive impairment due to sleep deprivation is conjoined with mood disorders and particularly symptom severity [47]. Emotional information is misinterpreted, making sleep-deprived subjects prone to anxiety [48] and depressive symptoms [49] as well as altered reward-seeking and impulsive behaviors [50]. Stress is one of the main factors influencing sleep. Hyperarousal is a key component in all modern etiological models of insomnia disorder. Overactive neurobiological and psychological systems contribute to sleep onset disorders. Sleep reactivity is the degree to which stress disrupts sleep, manifesting as difficulty falling and staying asleep. Individuals with highly reactive sleep systems experience drastic deterioration of sleep when stressed, whereas those with low sleep reactivity proceed largely unperturbed during stress. Research points to genetics, family history of insomnia, gender, and environmental stress as factors that influence sleep reactivity. High sleep reactivity is also linked to risk of shift-work disorder, depression, and anxiety [51,52,53] (Figure 2).

Exercise could improve to one extent cognitive performance. High-intensity resistance training has shown to enhance memory and critical thinking while ameliorating the symptomatology of mood disorders [54]. Concomitantly, aerobic exercise prevented further cognitive deterioration in cases of mild cognitive impairment [55]. The advancement in understanding and implementing exercise in patients with underlying pathology has supplemented training programs for professional athletes with techniques to reinforce cognitive performance along with athletic [56]. Due to the great variety of sports, there are different requirements that presuppose the existence of individualized programs. Therefore, future studies could focus on specific groups of athletes and highlight personalized programs centered on sleep hygiene.

### 2.3. Sleep Deprivation and Exercise: Cardiovascular Implications—The Example of Sleep Apnea

Recent epidemiological studies have revealed relationships between sleep deprivation and hypertension, coronary heart disease, and diabetes mellitus due to increased activity of sympathetic system [57,58]. Obstructive sleep apnea–hypopnoea syndrome (OSAHS) is associated with increased cardiovascular morbidity and mortality. Subjects with OSAHS and no known cardiovascular disease had increased arterial stiffness and impaired endothelial function compared to controls [34] (Table 1). A brief, mild hypercapnic exposure increases vascular resistance in the renal and segmental arteries [32]. Sleep-disordered breathing, short sleep time, and low sleep quality are frequently reported by patients with heart failure (HF). Sleep-disordered breathing, which includes OSA and CSA, is common in patients with HF and has been suggested to increase the morbidity and mortality in these patients. Both OSA and CSA are associated with increased sympathetic activation, vagal withdrawal, altered hemodynamic loading conditions, and hypoxemia [59].

There are several parameters that describe the mechanism that leads to increased risk of cardiovascular impairment. Sleep-disordered breathing, such as in OSA, activates the sympathetic system and contributes to systemic inflammation, metabolic dysregulation, vascular endothelial dysfunction, and uncoupling of myocardial workload [5,7]. Moreover, high blood pressure and increased heart rate combined with increased oxygen demand, accompanying hypertension and dyslipidemia, lead to variety of cardiovascular diseases, such as atherosclerosis and even heart failure [5,7]. Chronic sleep deprivation is associated with increased risk of cardiometabolic disease (Figure 2). Laboratory studies demonstrate that sleep deprivation causes impaired whole-body insulin sensitivity and glucose disposal. Evidence suggests that inadequate sleep also impairs adipose tissue insulin sensitivity and the NEFA rebound during intravenous glucose-tolerance tests [60]. In addition, muscle recovery is hindered when athletes are sleep deprived through inflammatory exacerbation [61].

In conclusion, potential mechanisms of influence on quality and quantity of sleep may allow scientists to positively influence sleep in athletes and maximize their performance and health [41]. Exercise itself may result in a fundamental therapeutic approach, as preliminary data have shown that it restabilizes sleep architecture and quality [62]. In fact, some novel therapeutic strategies have emerged related to inspiratory muscle training. Inspiratory muscle strength training (IMT) has shown promising results in managing both sleep apneas and arterial hypertension [63,64]. Assessing and training inspiratory strength in athletes could prove beneficial in counteracting the detrimental effects of the aforementioned sleep disturbances [65].

### 2.4. Sleep Deprivation and Performance

Sleep optimization via sleep extension has been shown to enhance athletic performance and provide increased benefits regarding aerobic function and metabolism [66]. Beneficial effects attributed to longer sleep periods have also been observed in basketball [67], handball [68], and rugby players [69], among others. Aside from general aspects of health and performance, sleep optimization has shown to improve specific aspects of the athlete’s performance, i.e., serve accuracy in tennis and stroke performance in swimming [42], as well as cognitive aspects [70], with a high dependency on the quality of sleep and its architecture [71]. Notably, sleep extension may be achieved by supplementing sleep with fixed naps, shown to significantly diminish sleep inertia and promote overall better performance [72].

Conversely, diminished sleep may be detrimental not only performance-wise but as a contributor to training and performance-related injuries [73]. A study in elite female football athletes has shown that there is significant inter-individual variability, and hence, personalized approaches in promoting sleep health should be adopted [74]. The latter concept can be generalized in several sports and with expert recommendations clearly advocating a case-based approach to sleep optimization [75].

## 3. Beneficial Sleep Effects

Sleep, in particular slow-wave sleep, is a restorative state that enables recovery from prior wakefulness and fatigue by repairing processes and repleting energy [76]. Sleep has also been shown to have a restorative effect on the immune system and the endocrine system, facilitate the recovery of the nervous system and metabolic cost of the waking state, and play an integral role in learning, memory and synaptic plasticity, all of which can impact both athletic recovery and performance [77]. Adequate sleep duration and consistency with its internal organization, namely four to ix NREM/REM cycles, each lasting approximately 90 to 110 min [78], as well as quality may be important for preventing cardiovascular diseases in modern society [58]. Even midday, short-term breaks of napping have been proved to be as valuable as extending nighttime sleep [79], in particular when combined with exercise [80]. Wakefulness results in an oxidative burden, and sleep provides a protective mechanism against these harmful effects [26].

## 4. Conclusions

Optimal sleep extends its benefits to all systems, exerting its main effect on cognition and the cardiovascular and respiratory system. Adequate sleep quality and quantity consist of the two components crucial for the effective human function and restitution. Conversely, sleep deprivation undermines these effects with significant declines in cognitive tasks and hindered cardiovascular adaptability and responses. It is possible that an extension of sleep duration could prevent these detrimental effects and enhance its efficient role.

## Figures and Tables

**Figure 1 jpm-12-00383-f001:**
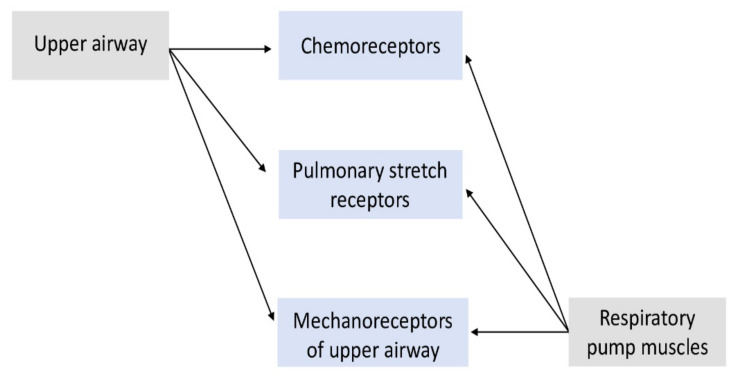
All the reflexes that take part in the control of respiratory rate during sleep. As inspiration occurs, upper airway muscles are activated by the mechanoreceptors, resulting in a protective reflex that prevents occlusion of airflow without arousals. However, inspiratory activation may become insufficient in terms of timing and magnitude due to stronger activation of respiratory pump muscles that lead to inadequate compensation for the airway-collapsing effect of negative inspiratory pressure.

**Figure 2 jpm-12-00383-f002:**
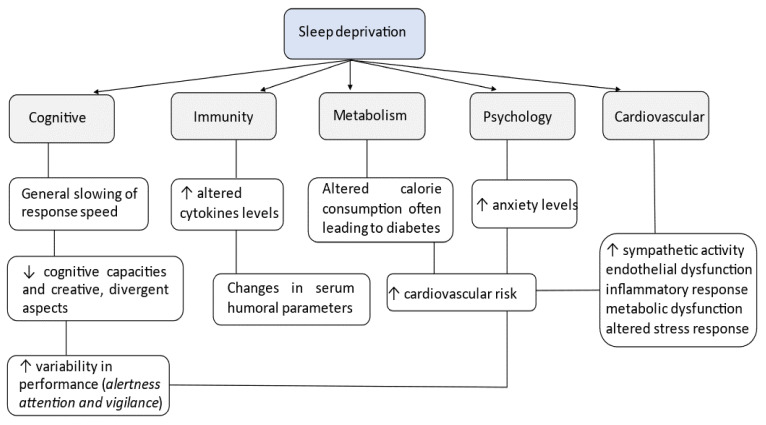
Sleep deprivation on general population.

**Table 1 jpm-12-00383-t001:** Responses and sleep deprivation.

References	Participants	Intervention Protocol	Results
Chapman et al. [32]	Healthy adults (age: 26.0 ± 4.0 yrs, M: *n* = 7, F: *n* = 7)	Blood velocity was measured in the renal and segmental arteries with Doppler ultrasound while subjects breathed room air and while they breathed a 3% CO_2_, 21% O_2_, 76% N_2_ gas mixture for 5 min	CO_2_ decreased blood velocity in the renal and segmental arteries and increased vascular resistance in the renal and segmental arteries (kidneys are hemodynamically responsive to a mild and acute hypercapnic stimulus in healthy humans)
Lei et al. [33]	14 healthy and right-handed adult males (mean age: 25.9 years) with normal or corrected-to-normal vision	fMRI study during RW and after 36 h of TSD	Self-reported scores of sleepiness were higher for TSD than for RW. A subsequent working memory task showed that memory performance was lower after 36 h of TSD. Significant increase of sleep pressure index was observed after 36 h of TSD
Van Eyck et al. [22]	120 children; control (age: 12.0 ± 3.0 y, M: *n* = 30, F: *n* = 55), mild OSAS (age: 11.0 ± 3.0 y, M: *n* = 9, F: *n* = 11), and moderate-to-severe OSAS (age: 12.0 ± 3.0 y, M: *n* = 10, F: *n* = 5)	PSG and a blood sample was taken to determine CRP levels	Relationship between CRP and BMI and between CRP and fat mass
Jones et al. [34]	OSAHS patients (age: 44.0 ± 7.0 y, M: *n* = 13, F: *n* = 7, AHI: ≥15/h and ESS score ≥ 11) vs. controls (age: 44.0 ± 7.0 y, M: *n* = 13, F: *n* = 7)	Evaluation of arterial stiffness (applanation tonometry and cardiovascular MRI) and endothelial function (measuring vascular reactivity after administration of glyceryl trinitrate and salbutamol)	Subjects with OSAHS had increased arterial stiffness and impaired endothelial function and were at increased risk for cardiovascular disease
Robertson et al. [35]	Healthy and normal-weight male students aged 20–30 y, BMI: 19–26 kg/m^2^. They were randomized to either sleep restriction (habitual bedtime minus 1.5 h) or a control condition (habitual bedtime) for three weeks	Weekly assessments of insulin sensitivity by hyperinsulinemic-euglycemic clamp, anthropometry, vascular function, leptin, and adiponectin were made. Sleep was assessed continuously using actigraphy and diaries.	Sleep restriction led to changes in insulin sensitivity, body weight, and plasma concentrations of leptin, which varied during the 3-week period. There was no effect on plasma adiponectin or vascular function. Even minor reductions in sleep duration led to changes in insulin sensitivity, body weight, and other metabolic parameters, which vary during the exposure period.

Abbreviations: AHI, apnea–hypopnea index; BMI, body mass index; CO_2_, carbon dioxide; CRP, C-reactive protein; ESS, Epworth Sleepiness Scale; F, female; fMRI, functional magnetic resonance imaging; M, male; *n*, number; N_2_, nitrogen; O_2_, oxygen; OSAHS, obstructive sleep apnea–hypopnea syndrome; OSAS, obstructive sleep apnea syndrome; PSG, polysomnography study; RW, rested wakefulness; TSD, total sleep deprivation.

## Data Availability

All data are available after request.

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
