# Peer review of "Elements of Sleep Breathing and Sleep-Deprivation Physiology in the Context of Athletic Performance"

_jpm, 2022, doi:10.3390/jpm12030383_

Round 1
Reviewer 1 Report
Thank you for the invitation to review this interesting paper.
It is generally well written, however the topic is ‘Elements of sleep breathing and sleep deprivation physiology in the context of athletic performance’.
I would recommend revise this paper and detailed information concerning the topic.
There are many interesting information about the physiology of breathing, however this paper contains too little information about the impact of exercises or physical activity on sleep breathing and deprivation.
Only a few sentences are written about sleep deprivations and exercises. I’d recommend revising ‘sleep deprivation and exercise’ to focus on the topic add details about the relationship of physical activity/exercises and sleep. Also, in this part of paper exercises are mixed with stress, anxiety and depression.
Sleep breathing disorders – please revise this part of the paper and add what is already known about sleep breathing in the context of athletic performance.
Line 170 Sleep deprivation of 30-to-72 hours – per week/month?
There are a few words that should be deleted, eg.
Line 20: ‘Among the general population, athletes suffer from these consequences more so during their performance’ – in my opinion ‘so’ should be deleted.
Line 164 – please remove “and’:
Conclusions:
“Adequate sleep quality and quantity” – what is the adequate sleep quality and quantity? Please add recommendation about is to the paper.
Author Response
We thank the reviewer for the comments that have helped us to improve the paper. All changes have been indicated by red color within the text. Below you will find a point-by-point response to your comments.
Reviewer 1
Thank you for the invitation to review this interesting paper. It is generally well written, however the topic is ‘Elements of sleep breathing and sleep deprivation physiology in the context of athletic performance’. I would recommend revise this paper and detailed information concerning the topic.
Comments 1. There are many interesting information about the physiology of breathing, however this paper contains too little information about the impact of exercises or physical activity on sleep breathing and deprivation.
Response: We thank the reviewer for his/her marking. We supplemented the manuscript with further details regarding the effects of exercise in the “Sleep deprivation and exercise” sections.
Comments 2. Only a few sentences are written about sleep deprivations and exercises. I’d recommend revising ‘sleep deprivation and exercise’ to focus on the topic add details about the relationship of physical activity/exercises and sleep. Also, in this part of paper exercises are mixed with stress, anxiety and depression.
Response: We thank the reviewer for his/her suggestion. We supplemented the section with additional data and references, along with clarifications, to complete the subject discussed.
Comments 2. Sleep breathing disorders – please revise this part of the paper and add what is already known about sleep breathing in the context of athletic performance.
Response: We thank the reviewer for his/her suggestion. We remodified the section and integrated it in the physiology implications. We, also, added a “sleep breathing and exercise” section, as proposed.
Comments 3. Line 170 Sleep deprivation of 30-to-72 hours – per week/month?
Response: We thank the reviewer for his/her highlighting. The paper cited was referring to sleep deprivation for 30-to-72 hours consecutively. We added a clarification in the sentence.
Comments 4. There are a few words that should be deleted, eg.
Response: We thank the reviewer for his/her remark. We proofread the manuscript and corrected any mistakes.
Comments 5. Line 20: ‘Among the general population, athletes suffer from these consequences more so during their performance’ – in my opinion ‘so’ should be deleted.
Response: We thank the reviewer for his/her suggestion. It was deleted.
Comments 6. Line 164 – please remove “and’:
Response: We thank the reviewer for his/her suggestion. It was deleted.
Comments 7. “Adequate sleep quality and quantity” – what is the adequate sleep quality and quantity? Please add recommendation about is to the paper.
Response: We thank the reviewer for his/her proposal. We supplemented the manuscript with additional details and citation regarding the definition for sleep quality and quantity.
Reviewer 2 Report
This is a very interesting paper.
The general impression is good, but this is a lack of information about articles about the influence of sleep restriction and longer sleep for performance in specific sports performance like basketball, tennis, etc. This lack of information should be corrected.
Author Response
We thank the reviewer for the comment that have helped us to improve the paper. All changes have been indicated by red color within the text. Below you will find a point-by-point response to your comments.
Reviewer 2
Comments 1. This is a very interesting paper. The general impression is good, but this is a lack of information about articles about the influence of sleep restriction and longer sleep for performance in specific sports performance like basketball, tennis, etc. This lack of information should be corrected.
Response: We thank the reviewer for his/her suggestion. It has been added more information at the section "Sleep deprivation and performance"
Reviewer 3 Report
Introduction
Neurocognitive performance, especially memory and learning, have been correlated with the presence of obstructive apneas. please cite doi:10.3390/bs11120180
Discussion
Sleep is an integral part of good health. Sleep disturbances and changes in sleep habits are associated with low-grade inflammation, which can be a cause or consequence of other conditions, including obesity, diabetes, and cardiovascular disease. Several strategies are available to counter these conditions, including continuous positive airway pressure (CPAP), pharmacological and nutritional interventions, and even surgery. Exercise and lowering BMI have shown promising results. please cite doi:10.1186/1476-511X-10-148
Author Response
We thank the reviewer for the comments that have helped us to improve the paper. All changes have been indicated by red color within the text. Below you will find a point-by-point response to your comments.
Reviewer 3
Comments 1. Introduction: Neurocognitive performance, especially memory and learning, have been correlated with the presence of obstructive apneas. please cite doi:10.3390/bs11120180
Response: We thank the reviewer for his/her suggestion. It was added.
Comments 2. Discussion: Sleep is an integral part of good health. Sleep disturbances and changes in sleep habits are associated with low-grade inflammation, which can be a cause or consequence of other conditions, including obesity, diabetes, and cardiovascular disease. Several strategies are available to counter these conditions, including continuous positive airway pressure (CPAP), pharmacological and nutritional interventions, and even surgery. Exercise and lowering BMI have shown promising results. please cite doi:10.1186/1476-511X-10-148
Response: We thank the reviewer for his/her suggestion. We supplemented discussion accordingly.
Round 2
Reviewer 1 Report
Thank you. The authors have already clarified all questions and suggestions. I accept the revised version.
Reviewer 2 Report
The manuscript is significantly improved can be accepted in its current form